# Cartilage Homeostasis under Physioxia

**DOI:** 10.3390/ijms25179398

**Published:** 2024-08-29

**Authors:** Yuji Arai, Ryota Cha, Shuji Nakagawa, Atsuo Inoue, Kei Nakamura, Kenji Takahashi

**Affiliations:** 1Department of Sports and Para-Sports Medicine, Graduate School of Medical Science, Kyoto Prefectural University of Medicine, Kawaramachi-Hirokoji, Kamigyo-ku, Kyoto 602-8566, Japan; shushi@koto.kpu-m.ac.jp; 2Department of Orthopaedics, Graduate School of Medical Science, Kyoto Prefectural University of Medicine, Kawaramachi-Hirokoji, Kamigyo-ku, Kyoto 602-8566, Japan; r-cha@koto.kpu-m.ac.jp (R.C.); a-inoue@koto.kpu-m.ac.jp (A.I.); keinaka2@koto.kpu-m.ac.jp (K.N.); t-keji@mbox.kyoto-inet.or.jp (K.T.)

**Keywords:** cartilage, chondrocytes, hypoxia, physioxia, HIF, osteoarthritis

## Abstract

Articular cartilage receives nutrients and oxygen from the synovial fluid to maintain homeostasis. However, compared to tissues with abundant blood flow, articular cartilage is exposed to a hypoxic environment (i.e., physioxia) and has an enhanced hypoxic stress response. Hypoxia-inducible factors (HIFs) play a pivotal role in this physioxic environment. In normoxic conditions, HIFs are downregulated, whereas in physioxic conditions, they are upregulated. The HIF-α family comprises three members: HIF-1α, HIF-2α, and HIF-3α. Each member has a distinct function in articular cartilage. In osteoarthritis, which is primarily caused by degeneration of articular cartilage, HIF-1α is upregulated in chondrocytes and is believed to protect articular cartilage by acting anabolically on it. Conversely, in contrast to HIF-1α, HIF-2α exerts a catabolic influence on articular cartilage. It may therefore be possible to develop a new treatment for OA by controlling the expression of HIF-1α and HIF-2α with drugs or by altering the oxygen environment in the joints.

## 1. Introduction

Osteoarthritis (OA) is the most prevalent joint disease, a degenerative condition primarily caused by the degeneration of articular cartilage. It is estimated that more than 250 million individuals worldwide are affected by this disease [1]. The symptoms of OA include pain and a limited range of motion. The loss of joint function caused by OA results in a decline in ADL, which in turn leads to a decrease in work capacity and, in the case of the elderly, the need for long-term care [2,3]. The primary approach is conservative therapy. Conventional pharmacological treatments, including oral acetaminophen and nonsteroidal anti-inflammatory drugs, as well as intra-articular injections of hyaluronic acid and corticosteroids, have been employed. In recent years, biotherapies, such as platelet-rich plasma (PRP) therapy and stem cell therapy, have also been employed to provide effective pain relief and functional improvement. However, these therapies are unable to control the degeneration of articular cartilage. Consequently, surgical treatments such as osteotomy or joint replacement are frequently selected when the stage of OA has progressed. Therefore, the reduction in ADL and the increased medical costs associated with OA represent a significant burden to countries around the world, underscoring the urgent need for the development of effective prevention and treatment strategies for OA [4]. The development of OA is associated with a number of factors, including age, gender, obesity, trauma, and rheumatoid arthritis [5]. At the molecular level, abnormal chondrocyte metabolism (decreased production of extracellular matrix components such as type II collagen and aggrecan), increased chondrocyte apoptosis, and destruction of the extracellular matrix by a number of proteolytic enzymes are all associated with OA. Each of these factors has been the subject of extensive research. However, the pathogenesis of OA is complex and not yet fully understood. Furthermore, the drug delivery system to chondrocytes has yet to be established, and, thus, an effective treatment for inhibiting the progression of OA has yet to be established [6]. In contrast to other tissues, articular cartilage is subjected to a markedly hypoxic environment (i.e., physioxia), and the transcription factor HIF-1α plays a pivotal role in maintaining articular cartilage homeostasis. Additionally, HIF-1α has been identified as a key factor in OA progression, and has recently been proposed as a potential therapeutic target for OA. This review presents a synthesis of the representative findings on articular cartilage and physioxia, as well as the most recent findings from approximately 50 articles searched in PubMed for the period between 2014 and 2024 and using the keywords “chondrocytes”, “osteoarthritis”, “hypoxia”, and “HIF”, with the exclusion of the keywords “mandibular cartilage” and “plant-derived products”.

## 2. Structure and Regulation of HIF

Oxygen and nutrients are essential for the survival of cells and organs in the body, and are primarily transported by the bloodstream. However, there are also non-vascularized tissues in the body, such as the cornea and intervertebral discs, which serve as representative examples of articular cartilage. The wet weight of articular cartilage is 50–85% water, with the remainder composed of an extracellular matrix (type II collagen and aggrecan) and a small number of chondrocytes. Despite receiving nutrients and oxygen from the synovial fluid to maintain homeostasis, articular cartilage is exposed to a physioxic environment and exhibits a superior hypoxic stress response compared to tissues with abundant blood flow. The oxygen concentration in the synovial fluid within joints is typically maintained at 6–9%, while the oxygen concentration in the deeper layers of articular cartilage is reduced to approximately 1% (Figure 1) [7]. A reduction in oxygen levels in eukaryotic cells results in the inhibition of oxygen-dependent mitochondrial respiration, with ATP production occurring via the glycolytic system. Therefore, the predominant energy metabolism in chondrocytes is anaerobic glycolysis with minimal oxidative phosphorylation [8]. Hypoxia-inducible factor (HIF) plays a pivotal role in this hypoxic environment, inducing hypoxia-related genes and regulating the cellular oxygen environment. In the 1990s, Gregg Semenza identified a protein that activates the erythropoietin gene during hypoxia in the hepatocellular carcinoma cell line Hep3B and designated it “HIF” [9]. Additionally, Peter Ratcliffe and William Kaelin Jr. elucidated a molecular mechanism that regulates hypoxia response genes in their response to oxygen concentration [10,11]. In 1995, HIF was identified as a transcription factor comprising a heterodimer of two distinct subunits (α and β) that are repressed in normoxic conditions and upregulated in hypoxic environments [12]. The HIF-α family comprises three distinct subunits, namely HIF-1α, HIF-2α, and HIF-3α, which exhibit specific functions in articular cartilage. HIF-1α is a ubiquitously expressed protein consisting of 826 amino acids, and is encoded by the HIF-1α gene, which is located on chromosome 14q21-24. The N-terminal side of HIF-1α contains a basic helix-loop-helix (bHLH) region and a Per-ARNT-Sim homology (PAS) region, which are involved in DNA binding. In contrast, the C-terminal side contains an N-terminal transactivation domain (N-TAD) and a C-terminal transactivation domain (C-TAD). At the core of HIF-1α is an oxygen-dependent degradation (ODD) domain, which serves as a sensor for oxygen concentration. In a normoxic environment, the two proline residues within this domain, Pro402 and Pro564, are known to undergo prolyl hydroxylation by prolyl hydroxylases (PHD). The hydroxylated proline residues are ubiquitinated by a ubiquitin ligase complex containing the tumor suppressor molecule von Hippel-Lindau tumor suppressor protein (VHL), resulting in rapid proteasome-dependent degradation of HIF-1α (Figure 2). Consequently, the intracellular accumulation of HIF-1α is markedly diminished. Another mechanism involves factor inhibiting HIF-1α (FIH-1), which, like PHD, requires oxygen for its activity. FIH-1 hydroxylates and modifies asparagine residues in the C-terminal transcriptional activation domain, thereby inhibiting binding to CBP/p300, a transcription-coupled factor. In hypoxic environments, however, these hydroxylation modifications are inhibited, allowing HIF-1α to accumulate in the cytoplasm, be phosphorylated, and translocated to the nucleus. There, it forms heterodimers with HIF-β/ARNT and binds to HREs of hypoxia-sensitive genes that are important for maintaining oxygen homeostasis, regulating energy metabolism, and promoting angiogenesis. Additionally, the genes involved in vasomotor control, apoptosis, proliferation, and extracellular matrix production (e.g., bone morphogenetic protein 2 (BMP2), BCL2, and adenovirus E1B 19-kDa-interacting protein 3 (BNIP3), erythropoietin (EPO), glucose transporter 1 (GLUT1), heme oxygenase-1 (HMOX-1), inducible nitric oxide synthase (iNOS), matrix metalloproteinases (MMPs), SOX-9, and vascular endothelial growth factor (VEGF)) are also affected [13] (Figure 3). HIF-1α is also known to be regulated by signaling pathways, including the PI3K/AKT and MAPK pathways [14]. As previously stated, the transcriptional activity of HIF is suppressed to minimal levels under normoxic conditions, despite the fact that HIF-1α is biosynthesized by the oxygen demand-dependent mechanisms of PHD and FIH-1. HIF-1α has been shown to have multiple functions in diseases of the kidney, heart, lung, gastrointestinal tract and central nervous system. For example, in renal anemia, it improves oxygen-carrying capacity by increasing red blood cell count, while in ischemic heart disease, it improves myocardial blood supply by inducing collateral angiogenesis. On the other hand, in cancer, it has the disadvantageous function of promoting angiogenesis, invasion, and metastasis. In recent years, a number of drugs that regulate HIF-1α have been investigated for the treatment of disease, including a recently approved drug that inhibits PHD and activates HIF-1α for the treatment of renal anemia. In addition, a drug that inhibits HIF-2α has been approved for the treatment of renal cell carcinoma. In the future, therapeutic agents that can control HIF-1α and HIF-2α are expected to be developed for OA [15].

## 3. HIF-1α in Normal Cartilage and Osteoarthritis

In normal cartilage, HIF-1α has been observed to promote glucose transport under physioxic conditions and to facilitate energy production through anaerobic glycolysis, which has been demonstrated to be anabolic for articular cartilage. HIF-1α also promotes the expression of SOX-9—which plays an essential role in chondrocyte differentiation—and of type II collagen and aggrecan, the typical extracellular matrix of chondrocytes, while suppressing the expression of ADAMTS-5 and MMP-13, which are typical proteolytic enzymes. Additionally, it activates autophagy and inhibits apoptosis [15]. These functions enable articular cartilage to adapt to physioxic environments and maintain homeostasis (Figure 4). Conversely, osteoarthritis (OA) is primarily a consequence of the deterioration of articular cartilage, yet it also impacts all constituent tissues of the joint. In other words, not only articular cartilage but also synovium, subchondral bone, and ligaments are affected, and HIF-1α expression plays a role in these changes. In osteoarthritis (OA), the oxygen concentration in the joint fluid is reduced due to increased oxygen consumption, which is caused by synovitis. This results in a reduction in oxygen levels within the articular cartilage, which is accompanied by an increase in HIF-1α expression within the OA articular cartilage. Yudoh et al. conducted an analysis of the expression pattern and role of HIF-1α in articular cartilage in OA. Their findings revealed that HIF-1α mRNA was highly expressed in degenerated areas in human OA cartilage samples, and that its expression level correlated with the degree of degeneration. The expression of HIF-1α mRNA and protein was found to be upregulated when cultured chondrocytes under physioxic conditions were stimulated with oxidative stress or IL-1β, which are catabolic factors associated with OA. This suggests that HIF-1α may play a role in the progression of OA. Furthermore, the placement of HIF-1α knockout cultured chondrocytes under physioxic conditions resulted in a reduction in energy production and extracellular matrix production, as well as an acceleration of stress-induced apoptosis [16]. The function of HIF-1α in chondrocytes in OA has been examined in animal models utilizing the HIF inhibitor 2-methoxyestradiol (2ME) and the PHD inhibitor dimethyloxaloylglycine (DMOG). Gelse et al. assessed the impact of intra-articular administration of 2ME and intra-articular administration of DMOG on OA progression in a murine model of OA. Their findings indicated that DMOG was ineffective in preventing the progression to severe OA, whereas 2ME was capable of inducing OA [17]. Hu et al. demonstrated that HIF-1α expression was elevated in articular chondrocytes from cases of human knee OA and in articular cartilage from mouse models of OA. Furthermore, they showed that DMOG augmented the chondroprotective effect, and that this was mediated by the anti-apoptotic effect of HIF-1α via mitophagy in chondrocytes [18]. Given the significant impact of polymorphic variants on the transcriptional activity of HIF1A, the gene encoding HIF-1α, Fernández-Torres et al. conducted a study to examine the prevalence of genetic polymorphisms in HIF1A among individuals with knee osteoarthritis (OA) in Mexico. The results indicated that the rs11549465 SNP may be implicated in the protection of articular cartilage, suggesting that HIF-1α plays a pivotal role in maintaining articular cartilage homeostasis [19]. Subsequently, Zhang et al. observed that HIF-1α expression was elevated in the synovium of knee joints in an animal model of OA. Furthermore, they found that HIF-1α suppression resulted in reduced synovial fibrosis and that HIF-1α knockdown in fibroblast-like synovial cells, which mimic the inflammatory environment of OA, led to decreased cell death and reduced expression of pyroptosis-related proteins. These findings suggest that increased HIF-1α in the synovium of knee OA may contribute to synovial fibrosis [20]. Moreover, aberrant bone remodeling in the subchondral bone, typified by hyperactive osteoclasts, is implicated in the progression of OA. Zhang et al. suppressed osteoclasts in the subchondral bone by knocking out lymphocyte cytosolic protein 1 (Lcp1) in a mouse OA animal model, which reduced bone remodeling in the subchondral bone and suppressed cartilage degeneration. Additionally, the knockout of Lcp1 inhibited angiogenesis by osteoclasts, maintained a physioxic environment in joints, and suppressed OA progression [21]. Li et al. collected subchondral bone samples from human OA and evaluated HIF-1α expression. The researchers demonstrated that HIF-1α expression was markedly elevated in severely sclerotic subchondral bone in comparison to less damaged subchondral bone. Furthermore, when primary human chondrocytes were cultured in physioxic conditions, matrix metalloproteinase 13 (MMP13) and MMP3 were significantly induced, while the mRNA expression of type II collagen, aggrecan, and SOX9 was suppressed. This suggests that hypoxia in the subchondral bone acts as a communication bridge between chondrocytes and osteoblasts [22].

## 4. The Function of HIF-1α in Chondrocytes

The HIF-α family comprises three members: HIF-1α, HIF-2α, and HIF-3α. Each member has distinct functions in articular cartilage [23,24]. HIF-1α is responsible for the production of energy through anaerobic glycolysis in normal cartilage. Furthermore, HIF-1α has been demonstrated to enhance glycolytic metabolism, including glucose uptake and ATP production, by regulating Runx2 in human OA chondrocytes [25]. HIF-1α exerts anabolic effects by promoting the expression of type II collagen and aggrecan, which are integral components of the extracellular matrix in articular cartilage, in chondrocytes within a physioxic milieu [26]. The forced expression of HIF-1α in chondrocytes, achieved through the use of cobalt chloride or gene transfer, has been observed to enhance the expression of type II collagen and aggrecan [27]. Conversely, the expression of several matrix metalloproteinases (MMP1, MMP2, MMP3, MMP13) and aggrecanases (ADAMTS4, ADAMTS5) is repressed when human cultured chondrocytes are exposed to physioxic conditions [28]. HIF-1α safeguards cartilage by inhibiting catabolic function and promoting anabolic function. Yang et al. conducted a knockout experiment on pyruvate kinase M2 (PKM2) in OA cultured chondrocytes and tested the suppression of HIF-1α expression. The results demonstrated that SOX-9, type II collagen, aggrecan, and GUT1 expression in chondrocytes was suppressed, and extracellular matrix degeneration was promoted [29]. In a mouse model of OA in which HIF-1α was knocked out, Bouaziz et al. observed that MMP13 expression was upregulated via Wnt signaling, resulting in exacerbated cartilage destruction [30]. As a mechanism, Okada et al. conducted an analysis of HIF-1α knockout mice and identified C1qtnf3 as a downstream molecule of HIF-1α that acts via the suppression of NF-κβ [31]. Conversely, Zhu et al. demonstrated that although the ubiquitin ligase F-box/WD repeat-containing protein 7 (FBW7) suppressed HIF-1α in chondrocytes following IL-1β treatment, the expression of SOX9, type II collagen, and aggrecan was promoted to exert a cartilage-protective effect [32]. Additionally, Li et al. demonstrated that DNA repair and recombination protein 54L can mitigate the inflammatory response associated with OA, even when HIF-1α is similarly suppressed [33]. These findings may be related to the signaling pathway from HIF-1α to VEGF. Additionally, HIF-1α plays a role in the autophagy and apoptosis of chondrocytes, with numerous recent findings emerging in this area, including via microRNA analysis. Zhang et al. and Chen et al. demonstrated that HIF-1α and miR-146a facilitate autophagy by suppressing Bcl-2 in cultured chondrocytes exposed to physioxia. Additionally, they showed that miR-146a targets Traf6 and IRAK1, but not Smad4 [34,35]. Yang et al. demonstrated that miR-411 regulates HIF-1α to promote autophagy in cultured chondrocytes treated with IL-1β [36]. Tsuchida et al. demonstrated that the induction of HIF-1α in cultured house rabbit chondrocytes resulted in the inhibition of apoptosis via HSP70 [37]. Additionally, Velard et al. showed that adrenomedullin (AM), a 52 amino acid hormone peptide, and its 31 amino acid truncated form, AM [22,23,24,25,26,27,28,29,30,31,32,33,34,35,36,37,38,39,40,41,42,43,44,45,46,47,48,49,50,51,52], inhibited chondrocyte apoptosis by suppressing Fas receptors in vitro [38]. Recently, HIF-1α has also been identified as a key regulator of mitophagy, a selective degradation mechanism of mitochondria. Lu et al. demonstrated that HIF-1α-enhanced human chondrocytes cultured under physioxic conditions activated autophagy, while HIF-1α knockout resulted in IL-1β stimulation-induced activation of mitophagy markers (BNIP3, PINK1, and Parkin) and led to the activation of apoptosis [39]. The interaction between the circadian clock and the HIF pathway in cartilage has also been the subject of recent research. Bmal1, a gene that plays a central role in the circadian rhythm, has been shown to regulate the expression of clock-regulated genes in cartilage tissue and to be involved in HIF1α expression and function [40]. Ma et al. reported that the knockout of the Bmal1 gene in cultured chondrocytes resulted in the suppression of HIF1α and VEGF expression and the promotion of apoptosis [41]. It has recently been reported that the expression of LncHIFCA, a long non-coding RNA, is upregulated in knee osteoarthritis (OA). LncHIFCA has been shown to positively regulate HIF-1α and promote OA pathogenesis by activating the PI3K/AKT/mTOR signaling pathway. Furthermore, LncZFHX2 is positively regulated by HIF-1α and maintains extracellular matrix homeostasis by repairing DNA in chondrocytes. Additionally, circRNA-UBE2G1, a circular RNA, regulates the miR-373/HIF-1a axis and promotes progression in an LPS-induced OA cell model. It has been demonstrated that these specialized RNAs may be involved in the pathogenesis of OA by affecting HIF-1α [42,43,44]. HIF-1α is also involved in chondrogenic differentiation. Zhang et al. reported that HIF-1α promotes the expression of SOX9, which plays an essential role in chondrogenesis [45]. Furthermore, stabilization of HIF-1α has been shown to promote chondrogenic differentiation and inhibit hypertrophy in undifferentiated cells.

Activation of SOX9 via HIF-1α in a physioxic environment has been demonstrated to promote differentiation of mesenchymal cells into chondrocytes [46]. Henrionnet et al. cultured human bone marrow mesenchymal stem cells (MSCs) in a medium containing TGFβ1 in a physioxic environment with alginate beads in 3D. The researchers observed that the expression of osteogenic markers was suppressed and chondrogenic markers were enhanced in cells cultured in this environment in comparison to cells cultured in a normoxic environment. Based on these findings, the researchers concluded that the physioxic environment is crucial for cartilage regeneration using MSCs [47]. Theodoridis et al. incorporated adipose-derived mesenchymal stem cells (ADMSCs) into 3D-printed honeycomb-like polycaprolactone (PCL) matrices and cultured them under different oxygen environments, with or without TGF. The findings indicated that the presence of TGFβ1 facilitated the differentiation of stem cells under physioxic conditions, resulting in the formation of hyaline cartilage with enhanced mechanical strength. The results demonstrated that, in addition to physioxia, the presence of TGF is essential for the successful generation of cartilage tissue with excellent biomechanical properties [48]. Conversely, it has been demonstrated that MSCs may undergo hypertrophic differentiation when subjected to prolonged culture in the presence of TGFβ. In a separate study, Li et al. discovered that MSCs were encapsulated in hyaluronic acid (HA) hydrogel and cultivated in the presence of TGF-β3. Subsequently, TGF-β3 was removed, and the MSCs were cultured under hypoxic conditions with mechanical stimulation to promote chondrogenic differentiation while inhibiting hypertrophic differentiation [49]. Additionally, Feng et al. utilized cartilage progenitor cells (CPCs) within gelatin methacryloyl microspheres (CGM) and observed that CPCs cultured in vitro under physioxic conditions effectively promoted chondrocyte proliferation and demonstrated robust chondrogenic differentiation potential compared to normoxic 3D CPCs and 2D cultured CPCs. Moreover, the in vivo injection of physioxic cultured CGMs into the joints of rat OA animal models demonstrated the capacity to suppress cartilage degeneration by inhibiting oxidative stress and inflammation. This suggests that this physioxic culture system may prove an effective regenerative medicine for OA [50]. Furthermore, recent evidence suggests that culturing iPS cells in a physioxic environment enhances HIF-1α expression and promotes chondrogenesis, a process that may be applied in regenerative medicine as a rapid and straightforward method for producing cartilage tissue [51].

## 5. The Treatment of Osteoarthritis (OA) with Hypoxia-Inducible Factor 1 Alpha (HIF-1α) Control

A variety of pharmacological agents, including acetaminophen and nonsteroidal anti-inflammatory drugs (NSAIDs), have been employed as a form of drug therapy for knee OA. The objective of this therapeutic approach is to provide pain relief, maintain or improve joint function, and enhance physical function. Nevertheless, disease-modifying osteoarthritis drugs (DMOADs), which have the potential to alter the course of OA, are still in the development phase and have yet to be implemented clinically. Conversely, fundamental research into the regulation of HIF1α has resulted in the creation of new therapeutic agents for OA, with numerous drugs and stimulation methods having been documented. In a mouse model of osteoarthritis (OA), Qin et al. demonstrated that resveratrol administered directly into the knee joints promoted autophagy in chondrocytes by regulating hypoxia-inducible factor 1 alpha (HIF-1α) and HIF-2α, thereby inhibiting articular cartilage degeneration [52]. Guan et al. discovered that capsiate, a gut microbiota metabolite, inhibited the expression of HIF-1α by activating SLC2A1 in chondrocytes, thereby preventing the progression of osteoarthritis (OA) associated with ferroptosis, a form of iron-dependent cell death [53]. Hyaluronic acid has been extensively utilized as a conventional intra-articular injectable formulation for knee osteoarthritis (OA), with numerous documented cases of its clinical efficacy. Its mechanism of action is not limited to lubrication; it also exhibits biological effects, including the enhancement of cartilage metabolism and the inhibition of proteolytic enzymes and chondrocyte apoptosis. HIF-1α has been demonstrated to augment the effects of hyaluronic acid. Ichimaru et al. have reported that the activation of HIF-1α in cultured chondrocytes under physioxic conditions increased the expression of the hyaluronic acid receptor CD44, which enhances the binding affinity to administered hyaluronic acid, thereby promoting cartilage metabolism [54]. Recently, intra-articular administration of autologous blood-derived PRP, which contains many growth factors, has been shown to promote cartilage repair in animal models and may improve symptoms such as joint pain in patients with knee osteoarthritis through anti-inflammatory effects. However, there is still a paucity of high-level evidence because the study design makes blinding difficult due to the autologous biomaterials, and there are variations in adjustment methods, such as platelet and white blood cell counts, between studies [55]. Moussa et al. observed that the addition of PRP to human OA chondrocytes increased autophagy and its markers BECLIN and LC3II, as well as HIF-1α expression, and decreased apoptosis [56]. Another important conservative treatment for knee OA is exercise therapy, which may be effective via the induction of HIF-1α in chondrocytes. Normoxic environments and moderate mechanical stress have been shown to induce HSP70 expression, improve cartilage metabolism, and inhibit the expression of degradative enzymes and apoptosis [57,58]. Shimomura et al. demonstrated that mechanical stress under physioxic conditions further enhances HIF-1α expression in cultured chondrocytes, while aggrecan expression is increased and ADATS5 expression is suppressed [59].

## 6. The Function of HIF-2α in Chondrocytes and the Treatment of OA by HIF-2α Regulation

HIF-2α, which was first identified in 1997, is an 870 amino acid protein that was initially demonstrated to be expressed in normal and human OA chondrocytes by Coimbra et al. [60]. HIF-2α has a comparable structure to HIF-1α and a comparable oxygen concentration-dependent regulatory mechanism. However, its function in articular cartilage is catabolic, in contrast to HIF-1α. The switching between HIF-1α and HIF-2α may serve as a potential mechanism for regulating the degeneration of articular cartilage. HIF-2α plays a role in endochondral ossification, promoting the expression of COL10A1, RUNX2, MMP13, and VEGF [61]. However, in articular chondrocytes, HIF-2α strongly induces the expression of these endochondral ossifying factors and is involved in the pathogenesis of OA. Saito et al. and Yang et al. have demonstrated that HIF-2α is involved in the pathogenesis and progression of OA via endochondral ossification signaling. This is evidenced by the fact that HIF-2α gene deletion suppresses OA, while HIF-2α gene transfer induces progressive joint destruction [61,62]. Furthermore, it has been shown that NF-kB is regulated upstream of HIF-2α [63]. Bohensky et al. demonstrated that HIF-2α is a potent regulator of the autophagy-promoting function of HIF-1α in mature chondrocytes [64]. Ryu et al. showed that HIF-2α expression is markedly upregulated in human and mouse OA chondrocytes in association with increased apoptosis and that it plays a role in articular cartilage destruction by significantly increasing chondrocyte apoptosis under agonistic anti-Fas antibody, which binds to FAS, a cell surface receptor [65]. Oh et al. and Yang et al. have demonstrated that HIF-2α stimulates NAD synthesis by directly targeting NAMPT, which has nicotinamide phosphoribosyltransferase activity, and activates the SIRT family, thereby destroying articular cartilage by acting on catabolic factors such as MMP3, MMP12, and MMP13 [66,67]. Lee et al. have demonstrated that HIF-2α activates and interacts with the zinc-ZIP8-MTF1 axis, thereby contributing to joint destruction [68]. Yang et al. have also implicated HIF-2α in joint destruction by targeting the multifunctional protein prokineticin 2 (Prok2) [69]. Furthermore, Yang et al. have reported that HIF-2α plays a role in the pathogenesis of OA through the Aurora kinase A (AURKA)/NEDD9 pathway, which results in the loss of primary cilia [70]. Additionally, HIF-2α expression is influenced by mechanical stress, similar to HIF-1α. Inoue et al. demonstrated that hydrostatic pressure applied to cultured chondrocytes resulted in HIF-2α expression and an increase in the expression of catabolic factors, including MMP3 and MMP13 [71]. Furthermore, Wang et al. investigated the effects of whole-body vibration (WBV) on a rat model of OA. Their findings indicated that low-frequency WBV reduced the expression of HIF-2α and catabolic factors, and inhibited articular cartilage degeneration. This may be attributed to mechanical stress [72]. As a result, several therapeutic approaches targeting HIF-2α have been developed. HIF-2α siRNA is delivered using biomaterials, such as nanoparticles and pH-responsive metal-organic frameworks, to regulate the gene expression of HIF-2α in articular cartilage. This is achieved through articular cartilage-specific effects and sustained release effects [73,74]. Furthermore, additional compounds have been identified as potential therapeutic agents, including resveratrol, osteopontin, chondromodulin 1 (an endogenous antiangiogenic protein of cartilage), D-mannose, and rhodanine derivatives [75,76,77,78,79]. Additionally, research has indicated that the therapeutic mechanism of PRP therapy in osteoarthritis (OA) is not solely reliant on the anti-apoptotic effects of HIF-1α, but also on the inhibition of HIF-2α [80]. Recently, microRNAs such as miR-365, miR-96-5p and -3P, and miR-96-5p have been demonstrated to inhibit HIF-2α and suppress cartilage matrix degeneration [81,82,83].

## 7. The Function of HIF-3α in Chondrocytes

The function of HIF-3α in chondrocytes has been the subject of considerable research since its identification in 1998 [84]. HIF-3α is a 662 amino acid protein. The structure of HIF-3α differs from that of HIF-1α and HIF-2α. It is hypothesised that HIF-3α represses the expression of HIF-1α and regulates the expression of chondrocyte hypertrophic differentiation genes, such as COL10A and MMP13, which are involved in HIF-2α [85,86,87]. However, the precise function of HIF-3α in chondrocytes remains unclear.

## 8. Regulatory Mechanisms of HIF-1α by Duration of Physioxia

The regulatory mechanisms of HIF-1α by duration of physioxia have been the subject of considerable research. There are two forms of cellular physioxic environments: chronic or sustained physioxia, when cells experience physioxia in a quasi-steady state, and intermittent, acute, transient, or cyclic physioxia, when cells fluctuate in time above and below the physioxic threshold. The behavior of cells in physioxic environments depends on the duration of the state [88]. The expression of intracellular HIF-1α is dependent on the duration of the physioxic environment. Kaihara et al. [89] demonstrated that culturing synovial cells or rearing rats in a sustained physioxic environment suppressed the expression of intracellular HIF-1α. Similarly, Kamada et al. [90] reported that rearing rats in sustained physioxia suppressed HIF-1α expression in muscle. As a mechanism, the protein expression of intracellular HIF-1α increases within a short period of time from normoxia to physioxia, generally peaking at 4 to 8 h. However, when physioxia is sustained for an extended period, PHD is activated by negative feedback and intracellular HIF-1α protein is degraded [91]. Furthermore, Li et al. observed that rearing rat OA models in a sustained physioxic environment for four weeks resulted in a notable advancement of OA symptoms compared to OA model rats reared in a normoxic environment [92]. Conversely, studies utilizing tumor and vascular endothelial cells have indicated that repeated intermittent hypoxia facilitates more efficient HIF-1α gene expression and inhibits its degradation in comparison to sustained hypoxia [93,94,95]. The mechanism by which intermittent physioxic stimulation increases HIF-1α protein expression is a topic of ongoing research. Martinive et al. have proposed that stimulation of the mitochondrial respiratory chain (increased oxygen consumption) induced by intermittent hypoxia and subsequent activation of the PI3K/Akt pathway during reoxygenation followed by hypoxia may contribute to increased expression of HIF-1α during stimulation [96]. Toffoli et al. have suggested that a continuous increase in PKA activity due to intermittent hypoxic stimulation may promote phosphorylation of HIF-1α during hypoxia and increase its transcriptional activity [97]. Additionally, it is established that reactive oxygen species (ROS) are produced at a higher rate during steady-state oxygenation following exposure to hypoxia than during hypoxia itself [98]. Malec et al. have demonstrated that ROS production enhanced by reoxygenation in intermittent hypoxia may serve as a regulator of antioxidant enzymes, including nuclear factor erythroid 2-related factor 2 (NRF2), a regulator of antioxidant enzymes, to enhance HIF-1α-related signaling and promote its stabilization [99]. Nevertheless, the precise mechanism by which intermittent physioxic stimulation enhances HIF-1α expression remains to be elucidated.

## 9. Conclusions

There is currently no curative drug therapy for osteoarthritis (OA), and the mainstay of management remains symptomatic treatment. Consequently, when OA progresses, arthroplasty is the primary surgical intervention, although it is not without complications in terms of invasiveness and cost. Conversely, the etiology of OA is being elucidated at the molecular level, and HIF, which plays a pivotal role in the physioxic environment of articular cartilage, has been identified as a key factor in the degeneration of articular cartilage, synovitis and subchondral bone remodeling in OA. It has become evident that there are distinct subtypes of HIF, including HIF-1α, HIF-2α, and HIF-3α. HIF-1α has been shown to mediate anabolic functions, whereas HIF-2α has been demonstrated to regulate catabolic processes. Furthermore, regulators of these factors have been identified, and methods to promote the expression of HIF-1α and inhibit the expression of HIF-2α have been studied and are anticipated to become disease-modifying agents for OA, as shown in Table 1 (Figure 5). Conversely, PHD inhibitors that inhibit and stabilize HIF-1α degradation have been employed as pharmacological agents for renal anemia [100], while HIF-2α inhibitors have been utilized in the treatment of renal cell carcinoma [101]. Further research on the roles of HIF-1α, HIF-2α, and HIF-3α and their methods of regulation may facilitate the development of new disease-modifying methods for OA.

## Figures and Tables

**Figure 1 ijms-25-09398-f001:**
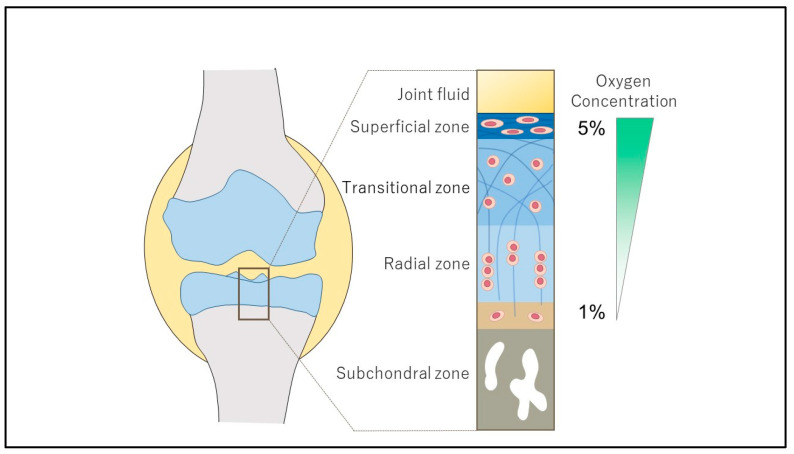
The structure of articular cartilage and the physioxic environment. Articular cartilage consists primarily of collagen fibers (blue string), water, and chondrocytes. From a histological perspective, the articular cartilage is composed of a lamina splendens, a superficial layer, an intermediate layer (transitional layer), a deep layer (radiating layer), and a calcified layer. Subchondral bone, which exhibits high bone density, is situated beneath the calcified layer. The oxygen environment within the synovial fluid within the joint is typically maintained at 6–9%, while the oxygen concentration within the articular cartilage decreases with depth, reaching approximately 1% at the deepest levels.

**Figure 2 ijms-25-09398-f002:**
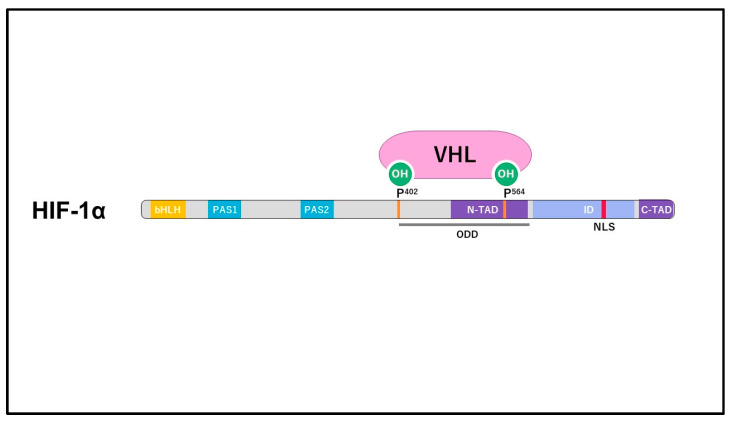
The structure of HIF-1α. It comprises a bHLH (basic helix–loop–helix) domain and a PAS (Per-ARNT-Sim homology) domain on the N-terminal side, and N-TAD (N-terminal transactivation domains) and C-TAD (C-terminal transactivation domains) on the C-terminal side. The ODD (oxygen-dependent degradation) domain region is situated in the middle of HIF-1α. In a normoxic environment, two proline residues, Pro402 and Pro564, undergo hydroxylation, and are subsequently ubiquitinated by VHL (von Hippel-Lindau tumor suppressor protein).

**Figure 3 ijms-25-09398-f003:**
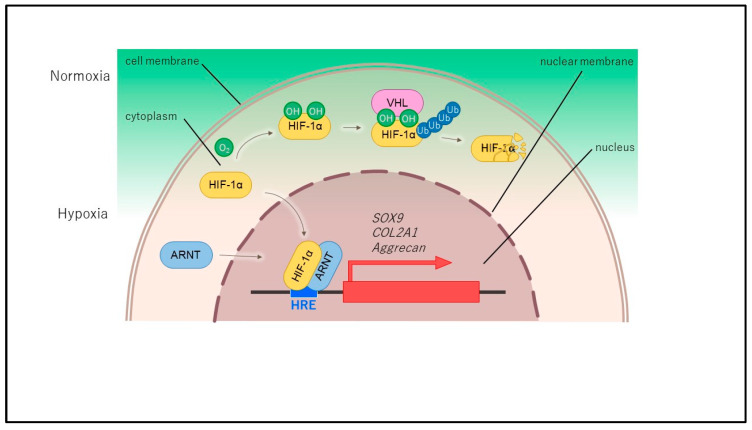
The mechanism of HIF1α activation in physioxic environments within articular cartilage is as follows. In physioxic conditions, HIF-1α accumulates in the cytoplasm due to the suppression of hydroxylation modification. It is then phosphorylated and translocated to the nucleus, where it forms a heterodimer with HIF-β/ARNT (Aryl hydrocarbon nuclear translocator). This complex binds to HREs (hypoxia response element) of hypoxia-sensitive genes, thereby activating gene transcription.

**Figure 4 ijms-25-09398-f004:**
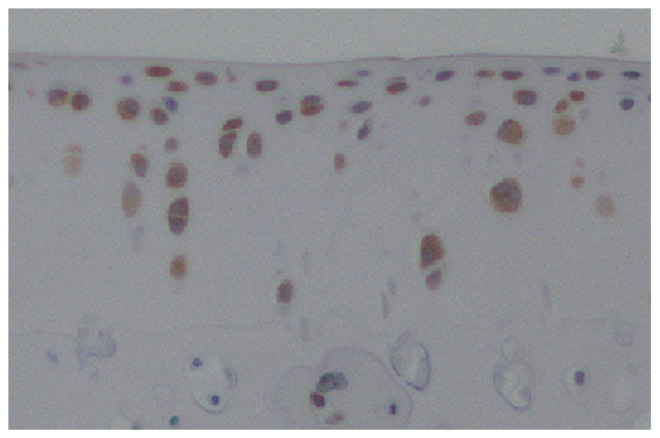
The expression of HIF1α in normal rat articular cartilage. HIF-1α is expressed in normal rat cartilage tissue during the process of adaptation to physioxic environments, with expression observed in both superficial and deep layers.

**Figure 5 ijms-25-09398-f005:**
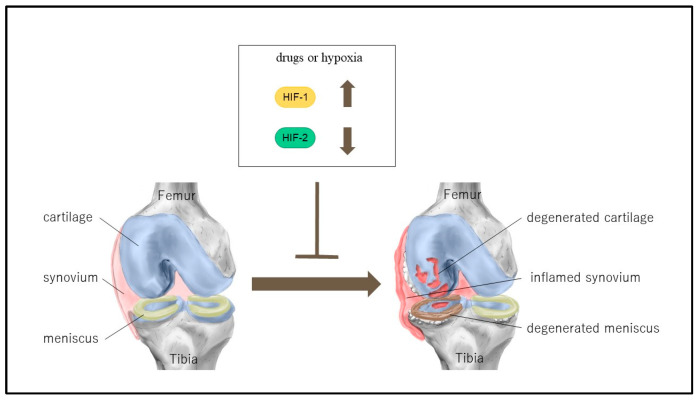
Treatment of osteoarthritis (OA) by controlling HIF-1α and HIF-2α. The treatment of osteoarthritis (OA) may be achieved by regulating hypoxia-inducible factor (HIF) expression through the use of pharmacological agents or by modulating the hypoxic environment. The protection of chondrocytes is achieved by the promotion of HIF-1α expression or the inhibition of HIF-2α expression, which can be accomplished through the use of pharmacological agents or by modulating the hypoxic environment.

**Table 1 ijms-25-09398-t001:** The potential of HIF regulation as a therapeutic avenue for OA.

Regulators	HIFs	Tissue (Cell)	Mechanism	References
DMOG	HIF-1α	mouse chondrocytes, mouse OA model	activation of mitophagy	Hu et al. (2020) [18]
FBW7	HIF-1α	human chondrocytes	maintenance of chondrogenic phenotype	Zhu et al. (2020) [32]
RAD54L	HIF-1α	human chondrocytes, rat OA model	anti-inflammation	Li et al. (2024) [33]
miR-146a	HIF-1α	mouse chondrocytes	activation of autophagy	Zhang et al. (2015), Chen et al. (2017) [34,35]
miR-411	HIF-1α	human chondrocytes	activation of autophagy	Yang et al. (2020) [36]
HSP70	HIF-1α	rabbit chondrocytes	inhibition of apoptosis	Tsuchida et al. (2014) [37]
AM	HIF-1α	bovine chondrocytes	inhibition of apoptosis	Velard et al. (2010) [38]
Bmal1	HIF-1α	mouse chondrocytes	inhibition of apoptosis	Ma et al. (2019) [41]
resveratrol	HIF-1α,2α	mouse OA model	activation of autophagy	Qin et al. (2017) [52]
capsiate	HIF-1α	mouse OA model	inhibition of ferroptosis	Guan et al. (2023) [53]
hyaluronic acid	HIF-1α	rat cartilage tissue	cartilage metabolism	Ichimaru et al. (2016) [54]
PRP	HIF-1α	human chondrocytes	activation of autophagy	Moussa et al. (2017) [56]
mechanical stimulation	HIF-1α	rat chondrocytes	cartilage homeostasis	Shimomura et al. (2021) [59]
WBV	HIF-2α	rat OA model	anti-inflammation, cartilage hometostasis	Wang et al. (2020) [72]
HIF-2α siRNA	HIF-2α	mouse OA model	anti-inflammation, cartilage regeneration	Pi et al. (2015), Zhang et al. (2023) [73,74]
resveratrol	HIF-2α	mouse OA model	inhibition of cartilage degradation	Li et al. (2015) [75]
Osteopontin	HIF-2α	human chondrocytes	CD44 interaction	Cheng et al. (2015) [76]
Chondromodulin-1	HIF-2α	rat OA model	cartilage homeostasis	Zhang et al. (2016) [77]
D-mannose	HIF-2α	mouse OA model	inhibition of ferroptosis	Zhou et al. (2021) [78]
rhodanine	HIF-2α	mouse OA model	inhibition of cartilage degradation	Kwak et al. (2022) [79]
PRP	HIF-2α	rat chondrocytes	inhibition of apoptosis, anti-inflammation	Yang et al. (2021) [80]
miR-365	HIF-2α	human chondrocytes	inhibition of catabolic factor	Hwang et al. (2017) [81]
miR-96-5p and -3P	HIF-2α	mouse OA model	cartilage homeostasis	Ito et al. (2021) [82]
miR-96-5p	HIF-2α	mouse OA model	cartilage homeostasis	Zhou et al. (2021) [83]

DMOG, dimethyloxaloylglycine; FBW7, F-box/WD repeat-containing protein 7; RAD54L, DNA repair/recombination protein 54 L; AM, adrenomedullin; PRP, platelet-rich plasma; WBV, whole-body vibration.

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
