# Peer review of "Cartilage Homeostasis under Physioxia"

_ijms, 2024, doi:10.3390/ijms25179398_

Round 1

Reviewer 1 Report

Comments and Suggestions for Authors

Huge review on cartilage homeostasis under physioxia.
In some cases, Physioxia should be employed.
Abstract: OK
Intro: line 44 "infections"? Instead, during RA, in my opinion.
Structure: OK
HIF 1: Please add a short paragraph on 3D culture and hypoxia  (chondrocytes, MSC-driven chondrogenesis, effects of growth factors, and influence on mineralization). I dislike citing plants, so please use only referenced molecules in Sigma.
The use of PRP therapy (like HA) is very debated in the clinics.
Is there any data on the influence of intermittent fasting on HIF, as observed on inflammation in mice (urate)?
HIF2: please define agnostic anti-Fas antibody (p8, line 330)
Regulatory mechanism (p9 l389): Please be precise about which rat OA model is used (e.g., MIA blocks anaerobic glycolysis, which is thus unadapted to this topic).
It would be beneficial for the authors to include a table summarizing the positive and negative effects of mediators, environment, and various drugs on HIFs. This will serve as a handy reference for the reader.

Author Response

I am grateful for your constructive and well-considered feedback on this review. In response to each of the points that you have raised, we have provided detailed responses. In the revised manuscript, additions and corrections are highlighted for clarity.

Comments 1: Huge review on cartilage homeostasis under physioxia. In some cases, Physioxia should be employed.

Response 1: I am grateful for the clarification regarding the terminology. As defined by McKeown and Hammond et al., the terms "normoxia," "physoxia/physioxia," and "hypoxia" are used in biology to describe the oxygen environment in which a tissue or cell is placed (Hammond EM, Asselin MC, Forster D, O'Connor JP, Senravich et al. “normoxia”, ‘physoxia/physioxia’ and ‘hypoxia’ depending on the oxygen environment in which the tissue or cell is placed (Hammond EM, Asselin MC, Forster D, O'Connor JP, Senra JM, Williams KJ. meaning, measurement and modification of hypoxia in the laboratory and the clinic. 2014, 26:277-288.) (McKeown SR. Defining normoxia, physoxia and hypoxia in tumours-implications for treatment response. Br J Radiol. 2014, 87(1035): 20130676). Many studies have been performed at atmospheric oxygen concentrations (20.9% Oâ‚‚), which is defined as "normoxia." Conversely, at the organ and tissue level, oxygen levels are reduced even under normal conditions, with levels varying depending on the tissue. However, even in a hypoxic environment, this environment is not defined as "normoxia" but rather as "physioxia." Conversely, an environment such as cancer tissue, where oxygen levels are diminished from the norm and oxygen supply to organs and tissues is insufficient, is defined as "hypoxia." In accordance with this definition, normal cartilage exists in a hypoxic environment and not in a pathological one. Consequently, the term "hypoxia" has been changed to "physioxia" in this review, including the title. However, we have also considered the historical background of the literature that uses the term "hypoxia" and the literature on tumor cells, and have decided to use the term "hypoxia" in this context.

Comments 2: Abstract: OK

Response 2: Thank you for your feedback.

Comments 3: Intro: line 44 "infections"? Instead, during RA, in my opinion.

Response 3: In response to the suggestion, the term "infections" has been replaced with "rheumatoid arthritis" in the revised manuscript.

Comments 4: Structure: OK

Response 4: Your input is appreciated

Comments 5: HIF 1: Please add a short paragraph on 3D culture and hypoxia (chondrocytes, MSC-driven chondrogenesis, effects of growth factors, and influence on mineralization).

Response 5: I have added the new sentences and the four new references on three-dimensional culture and hypoxia in the chapter on "The Function of HIF-1α in Chondrocytes."

Henrionnet et al. cultured human bone marrow mesenchymal stem cells (MSCs) in a me-dium containing TGFβ1 in a physioxic environment with alginate beads in 3D. The re-searchers observed that the expression of osteogenic markers was suppressed and chon-drogenic markers were enhanced in cells cultured in this environment in comparison to cells cultured in a normoxic environment. Based on these findings, the researchers con-cluded that the physioxic environment is crucial for cartilage regeneration using MSCs (47). Theodoridis et al. incorporated adipose-derived mesenchymal stem cells (ADMSCs) into 3D-printed honeycomb-like polycaprolactone (PCL) matrices and cultured them under different oxygen environments, with or without TGF. The findings indicated that the presence of TGFβ1 facilitated the differentiation of stem cells under physioxic conditions, resulting in the formation of hyaline cartilage with enhanced mechanical strength. The results demonstrated that, in addition to physioxia, the presence of TGF is essential for the successful generation of cartilage tissue with excellent biomechanical properties (48). Conversely, it has been demonstrated that MSCs may undergo hypertrophic differentiation when subjected to prolonged culture in the presence of TGFβ. In a separate study, Li et al. discovered that MSCs were encapsulated in hyaluronic acid (HA) hydrogel and cultivated in the presence of TGF-β3. Subsequently, TGF-β3 was removed, and the MSCs were cul-tured under hypoxic conditions with mechanical stimulation to promote chondrogenic differentiation while inhibiting hypertrophic differentiation (49). Additionally, Feng et al. utilized cartilage progenitor cells (CPCs) within gelatin methacryloyl microspheres (CGM) and observed that CPCs cultured in vitro under physioxic conditions effectively promoted chondrocyte proliferation and demonstrated robust chondrogenic differentiation potential compared to normoxic 3D CPCs and 2D cultured CPCs. Moreover, the in vivo injection of physioxic cultured CGMs into the joints of rat OA animal models demonstrated the capac-ity to suppress cartilage degeneration by inhibiting oxidative stress and inflammation. This suggests that this physioxic culture system may prove an effective regenerative medi-cine for OA (50).

  1. Henrionnet, C.; Liang, G.; Roeder, E.; Dossot, M.; Wang, H.; Magdalou, J.; Gillet, P.; Pinzano, A. Hypoxia for Mesenchymal Stem Cell Expansion and Differentiation: The Best Way for Enhancing TGFβ-Induced Chondrogenesis and Preventing Calcifi-cations in Alginate Beads. Tissue. Eng. Part A. 2017, 23: 913-922.
  2. Theodoridis, K.; Aggelidou, E.; Manthou, M.E.; Kritis, A. Hypoxia Promotes Cartilage Regeneration in Cell-Seeded 3D-Printed Bioscaffolds Cultured with a Bespoke 3D Culture Device. Int. J Mol. Sci. 2023, 24: 6040.
  3. Li, D.X; Ma, Z.; Szojka, A.R.; Lan, X.; Kunze, M.; Mulet-Sierra, A.; Westover, L.; Adesida, A.B. Non-hypertrophic chondrogen-esis of mesenchymal stem cells through mechano-hypoxia programing. J Tissue. Eng. 2023, 14:20417314231172574.
  4. Feng, K.; Yu, Y.; Chen, Z.; Wang, F.; Zhang, K.; Chen, H.; Xu, J.; Kang, Q. Injectable hypoxia-preconditioned cartilage progen-itor cells-laden GelMA microspheres system for enhanced osteoarthritis treatment. Mater. Today. Bio. 2023, 20: 100637.

Comments 6: I dislike citing plants, so please use only referenced molecules in Sigma.

Response 6: In response to the above comments, the following changes have been made: All references to plant-derived products in the context of HIF-1α and HIF-2α have been removed, and only the molecules and substances handled by Sigma are cited. The aforementioned information has been added at the end of the introduction section.

This review presents a synthesis of the representative findings on articular cartilage and physioxia, as well as the most recent findings from approximately 50 articles searched in PubMed between 2014 and 2024 using the keywords "chondrocytes," "osteoarthritis," "hypoxia," and "HIF," with the exclusion of the keyword "mandibular cartilage" and "plant-derived products."

Comments 7: The use of PRP therapy (like HA) is very debated in the clinics.

Response 7: In light of the paucity of evidence regarding the therapeutic effects and mechanisms of PRP, including the design of research studies and the methods of adjusting the numerous PRPs, the text on PRP has been revised as follows.

Recently, intra-articular administration of autologous blood-derived PRP, which contains many growth factors, has been shown to promote cartilage repair in animal models and may improve symptoms such as joint pain in patients with knee osteoarthritis through an-ti-inflammatory effects. However, there is still a paucity of high-level evidence because the study design makes blinding difficult due to the autologous biomaterials, and there are variations in adjustment methods, such as platelet and white blood cell counts, between studies (55).

  1. Oeding, J.F; Varad,y N.H; Fearington, F.W; Pareek, A.; Strickland, S.M; Nwachukwu, B.U; Camp, C.L; Krych, A.J. Platelet-rich plasma versus alternative injections for osteoarthritis of the knee: A systematic review and statistical fragility index-based meta-analysis of randomized controlled trials. Am. J Sports. Med. 2024, 29:3635465231224463.

Comments 8: Is there any data on the influence of intermittent fasting on HIF, as observed on inflammation in mice (urate)?

Response 8: Fasting therapy has been shown in animal and human clinical studies to be effective in the treatment of obesity, diabetes, cardiovascular disease, cancer, neurodegenerative disorders such as Alzheimer's disease and Parkinson's disease, Asthma, and Multiple Sclerosis. Fasting therapy also improves inflammation, as reported by Müller et al. (Müller H, de Toledo FW, Resch KL. Fasting followed by vegetarian diet in patients with rheumatoid arthritis: a Scand J Rheumatol. 2001, 30:1-10.). In addition, the effects of fasting on intracellular signaling pathways, such as Sirtuins-related factors, have been elucidated (de Cabo R, Mattson MP. Effects of Intermittent fasting on health, aging, and disease. N Engl J Med. 2019, 381:2541-2551.). However, the effect on HIF is currently unknown, and we chose not to mention the association between intermittent fasting therapy and HIF in this review.

Comments 9: HIF2: please define agnostic anti-Fas antibody (p8, line 330)

Response 9: As indicated, we have added the following definition:

agnostic anti-Fas antibody, which binds to FAS, a cell surface receptor (65).

Comments 10: Regulatory mechanism (p9 l389): Please be precise about which rat OA model is used (e.g., MIA blocks anaerobic glycolysis, which is thus unadapted to this topic).

Response 10: The data presented in this text were omitted as they related to an OA animal model of MIA.

Comments 11: It would be beneficial for the authors to include a table summarizing the positive and negative effects of mediators, environment, and various drugs on HIFs. This will serve as a handy reference for the reader.

Response 11: I have added a Table 1 as you indicated.

Table 1. The potential of HIF regulation as a therapeutic avenue for OA.

Reviewer 2 Report

Comments and Suggestions for Authors

Dear Authors, 

in the era of prosthesis implementation for OA, such as TKA and THA, the idea of focusing on cartilage homeostasis and specifically targeting signaling molecules with disease-modifying methods is good.

Indeed, the joint environment is increasingly studied in the literature, thanks to the advancement of technologies and the study of new signaling pathways.

Recent studies have focused on the role of hypoxia and the HIF-α family in controlling cartilage homeostasis.

You did a great job. Since this review summarized the possible targeting pathways and a parallel clinical mention, it could be accepted for publication after revision. 

Indeed, apart from the signaling molecules and transcription factors, it describes how the traditional treatment methods used for OA, such as hyaluronic acid, the new PRP and common exercise, could impact these factors.

However, there are some points that I want to share with you.

-​There are some sentences that should be removed since they are taken for granted as the first lines of the manuscript of the introduction, lines 23-26.

-​The bibliography should be updated.

-​Line 199: is there a mistake? (HIF-α family is the correct form instead of the HIF-1α family?)

- In lines 420 -421, the application of drugs targeting the HIF-α family is mentioned in the renal field, but not before. A brief introduction to existing drugs targeting these molecules employed in clinical practice should be introduced before.

-​Figure 5 is not so precise concerning promoting HIF-1alfa and inhibiting HIF-2alfa to avoid OA.

Author Response

I am grateful for your constructive and well-considered feedback on this review. In response to each of the points that you have raised, we have provided detailed responses. In the revised manuscript, additions and corrections are highlighted for clarity.

Comments 1: There are some sentences that should be removed since they are taken for granted as the first lines of the manuscript of the introduction, lines 23-26.

Response 1: In response to your feedback, the sentences and the reference have been removed.

Comments 2: The bibliography should be updated.

Response 2: In response to the comments provided by Reviewer 1, several references were removed and new references were added.

New references

  1. Yang, C.; Zhong, Z.F.; Wang, S.P.; Vong, C.T.; Yu, B.; Wang, Y.T. HIF-1: structure, biology and natural modulators. Chin. J Nat. Med. 2021, 19: 521-527.
  2. Henrionnet, C.; Liang, G.; Roeder, E.; Dossot, M.; Wang, H.; Magdalou, J.; Gillet, P.; Pinzano, A. Hypoxia for Mesenchymal Stem Cell Expansion and Differentiation: The Best Way for Enhancing TGFβ-Induced Chondrogenesis and Preventing Calcifi-cations in Alginate Beads. Tissue. Eng. Part A. 2017, 23: 913-922.
  3. Theodoridis, K.; Aggelidou, E.; Manthou, M.E.; Kritis, A. Hypoxia Promotes Cartilage Regeneration in Cell-Seeded 3D-Printed Bioscaffolds Cultured with a Bespoke 3D Culture Device. Int. J Mol. Sci. 2023, 24: 6040.
  4. Li, D.X; Ma, Z.; Szojka, A.R.; Lan, X.; Kunze, M.; Mulet-Sierra, A.; Westover, L.; Adesida, A.B. Non-hypertrophic chondrogen-esis of mesenchymal stem cells through mechano-hypoxia programing. J Tissue. Eng. 2023, 14:20417314231172574.
  5. Feng, K.; Yu, Y.; Chen, Z.; Wang, F.; Zhang, K.; Chen, H.; Xu, J.; Kang, Q. Injectable hypoxia-preconditioned cartilage progen-itor cells-laden GelMA microspheres system for enhanced osteoarthritis treatment. Mater. Today. Bio. 2023, 20: 100637.
  6. Oeding, J.F; Varad,y N.H; Fearington, F.W; Pareek, A.; Strickland, S.M; Nwachukwu, B.U; Camp, C.L; Krych, A.J. Platelet-rich plasma versus alternative injections for osteoarthritis of the knee: A systematic review and statistical fragility index-based meta-analysis of randomized controlled trials. Am. J Sports. Med. 2024, 29:3635465231224463.

Comments 3: Line 199: is there a mistake? (HIF-α family is the correct form instead of the HIF-1α family?)

Response 3: As indicated, this was a typographical error and has been corrected.

The HIF-α family comprises three members: HIF-1α, HIF-2α, and HIF-3α.

Comments 4: In lines 420 -421, the application of drugs targeting the HIF-α family is mentioned in the renal field, but not before. A brief introduction to existing drugs targeting these molecules employed in clinical practice should be introduced before.

Response 4: In order to assist readers in comprehending the pharmacological targeting of HIFs, as previously indicated, we have incorporated the following text and a new reference into the chapter entitled "Structure and Regulation of HIF," which now includes a detailed background section.

HIF-1α has been shown to have multiple functions in diseases of the kidney, heart, lung, gastrointestinal tract and central nervous system. For example, in renal anemia, it im-proves oxygen-carrying capacity by increasing red blood cell count, and in ischemic heart disease, it improves myocardial blood supply by inducing collateral angiogenesis. On the other hand, in cancer, it has the disadvantageous function of promoting angiogenesis, in-vasion and metastasis. In recent years, a number of drugs that regulate HIF-1α have been investigated for the treatment of disease, including a recently approved drug that inhibits PHD and activates HIF-1α for the treatment of renal anemia. In addition, a drug that in-hibits HIF-2α has been approved for the treatment of renal cell carcinoma. In the future, therapeutic agents that can control HIF-1α and HIF-2α are expected to be developed for OA (15).

  1. Yang, C.; Zhong, Z.F.; Wang, S.P.; Vong, C.T.; Yu, B.; Wang, Y.T. HIF-1: structure, biology and natural modulators. Chin. J Nat. Med. 2021, 19: 521-527.

Comments 5: Figure 5 is not so precise concerning promoting HIF-1alfa and inhibiting HIF-2alfa to avoid OA.

Response 5: As you pointed out, the figure was not readily comprehensible, so we have revised it to enhance its clarity.
